# Presence of Protease Inhibitor 9 and Granzyme B in Healthy and Pathological Human Corneas

**DOI:** 10.3390/biology11050793

**Published:** 2022-05-23

**Authors:** Stanislava Reinstein Merjava, Jan Kossl, Ales Neuwirth, Pavlina Skalicka, Zuzana Hlinomazova, Vladimir Holan, Katerina Jirsova

**Affiliations:** 1Laboratory of the Biology and Pathology of the Eye, Institute of Biology and Medical Genetics, First Faculty of Medicine, Charles University and General University Hospital in Prague, 128 00 Prague, Czech Republic; 2Department of Nanotoxicology and Molecular Epidemiology, Institute of Experimental Medicine of the Czech Academy of Sciences, 142 20 Prague, Czech Republic; kossl.jan@gmail.com (J.K.); vladimir.holan@iem.cas.cz (V.H.); 3Laboratory of Adaptive Immunity, Institute of Molecular Genetics of the Czech Academy of Sciences, 142 20 Prague, Czech Republic; ales.neuwirth@img.cas.cz; 4Department of Ophthalmology, General University Hospital in Prague and First Faculty of Medicine, Charles University, 128 08 Prague, Czech Republic; pavlina.skalicka@vfn.cz; 5Eye Clinic Lexum, 140 00 Prague, Czech Republic; hlinomazova@lexum.cz

**Keywords:** cornea, corneal endothelium, corneal epithelium, protease inhibitor 9, granzyme B

## Abstract

**Simple Summary:**

Detailed knowledge of the structure and properties of the human cornea is a prerequisite not only for the treatment of various corneal diseases but also for successful corneal transplantation and its long-term survival after grafting. Using various cell and molecular biology approaches, we found in cornea the protease inhibitor 9. This protein, known to be present in other human tissues but not yet reported in cornea, is directly involved in the immune response after transplantation. Together with its inhibitor (granzyme B), we localized this protein, especially in the superficial and inner cornea layers. This localization indicates that protease inhibitor 9 protein may be involved in protecting the cornea from external damage, but also in protection against immune cells inducing corneal graft rejection. Furthermore, we have shown on pathological corneal samples from corneal melting and herpes virus keratitis that the increased expression of both proteins is linked to these diseases. These experiments and their results represent an important contribution to the basic research of cornea biological properties with direct overlap into clinical practice.

**Abstract:**

The aim of this study was to find out whether protease inhibitor 9 (*PI-9*) and granzyme B (*GrB*) molecules that contribute to immune response and the immunological privilege of various tissues are expressed in healthy and pathological human corneas. Using cryosections, cell imprints of control corneoscleral discs, we showed that *PI-9* was expressed particularly in the endothelium, the superficial and suprabasal epithelium of healthy corneas, limbus, and conjunctiva. *GrB* was localized in healthy corneal and conjunctival epithelium, while the endothelium showed weak immunostaining. The expression of PI-6 and *GrB* was confirmed by qRT-PCR. Increased expression levels of the *PI-9* and *GrB* genes were determined when the corneas were cultured with proinflammatory cytokines. Fluorescent and enzymatic immunohistochemistry of pathological corneal explants (corneal melting and herpes virus keratitis) showed pronounced *PI-9*, *GrB*, human leucocyte antigen (HLA)-DR, and leukocyte-common antigen (CD45) signals localized in multicellular stromal infiltrates and inflammatory cells scattered in the corneal stroma. We conclude that increased expression of the *PI-9* and *GrB* proteins under pathological conditions and their upregulation in an inflammatory environment indicate their participation in immune response of the cornea during the inflammatory process.

## 1. Introduction

Protease inhibitor 9 (*PI-9*, serpin B9, Online Mendelian Inheritance in Man (OMIM) *601799) is a 42-kDa endogenous protein belonging to the serpins, the subfamily of serine protease inhibitors. Serpins regulate protease activity, play roles as chaperons or hormone transport proteins, and participate in numerous cellular processes, such as blood coagulation, inflammation, tissue remodeling, and apoptosis [1,2,3].

*PI-9* is mainly expressed by cytotoxic lymphocytes, antigen-presenting cells (dendritic and Langerhans cells), macrophages, and some B lymphocytes, and displays mostly nucleocytoplasmic localization [4,5,6,7]. Additionally, *PI-9* is present in the vascular endothelium, mesothelial and mast cells, and in various tissues at immune-privileged sites, such as the placenta, the ovary, and the testes [6,8,9,10]. *PI-9* is also highly expressed in various tumors [11]. The main function of *PI-9* is the specific inhibition of the activity of granzyme B (*GrB*, OMIM *123910) [5]. *GrB* is a caspase-like serine protease released by cytotoxic T lymphocytes and natural killer (NK) cells to destroy virus-infected cells or tumor cells via the activation of apoptosis [5]. It was shown that *GrB* plays an important role in transplant rejection and tumor immunosurveillance [12,13,14].

*PI-9* protects cells against the endogenous or inadvertent release of *GrB* and the premature or unwanted activation of the *GrB*-mediated cytotoxic cell death pathways [5,15]. At immune-privileged sites, *PI-9* may participate in protecting cells against *GrB* released from neighboring or circulating cytotoxic lymphocytes during an immune cytotoxic attack [6,9]. Similarly, the expression of *PI-9* in malignant cells is how they protect cells against self-inflicted damage [11].

In the human eye, only the retina and lens have been examined for the expression of *PI-9* to date, showing strong expression and immunostaining, respectively [6,16]. Based on this, we were interested in the localization of *PI-9* in the cornea, where various antigen-presenting cells participate in corneal immunity [17,18,19].

Additionally, we sought to analyze whether some changes in the expression of both *PI-9* and *GrB* proteins occur in pathological corneas with corneal melting (keratolysis) or herpes virus keratitis, where changes related to the immunological response might be expected [20,21].

## 2. Materials and Methods

### 2.1. Specimens

The study followed the standards of the Ethics Committee of the General University Hospital in Prague (GUH) and the First Faculty of Medicine, Charles University (FFMCU), and adhered to the tenets set out in the Declaration of Helsinki; informed consent was obtained from all participating subjects.

Twenty-four cadaverous corneoscleral discs (17 men and seven women; mean age, 60.8 ± 15.0 years) not acceptable for transplantation due to their low endothelial cell density were obtained from the Ocular Tissue Bank, Prague, or from the Institute of Pathology (GUH and FFMCU) and were used as controls. Five of the control buttons were dissected into four parts and snap-frozen in liquid nitrogen, embedded in an optimal cutting temperature compound, and stored at −80 °C. Tissue slices (7 μm thick) were prepared by cryosectioning and used for enzymatic immunohistochemistry. Six corneoscleral discs were used for impression cytology of the endothelium, epithelium, and conjunctiva on Biopore Millicell membranes (MILLICELL^®^ CM, PICM01250, Merck KGaA, Darmstadt, Germany) and subsequently used for indirect fluorescent immunocytochemistry. Thirteen corneoscleral discs were used for RNA isolation and quantitative reverse transcription PCR (qRT-PCR). Human tonsil and testes tissues served as positive controls for *PI-9* and *GrB*, respectively.

Additionally, five corneas stored under hypothermic conditions (mean storage time 5 days; four men and one woman; mean age 62 ± 5.4 years) were used for cultivation with the proinflammatory cytokines interferon (IFN)-γ and tumor necrosis factor (TNF)-α.

Finally, another 13 pathological corneal explants (nine men and four women; mean age, 55.2 ± 16.0 years) obtained from the GUH were used. Five corneal explants were obtained from patients with corneal or graft melting (non-infectious cases) and eight explants were obtained from patients with herpes virus keratitis.

### 2.2. Enzymatic Immunohistochemistry of Cryosections

Slices were stained with a single antibody as described previously [22] using a commercially available kit (Mouse and Rabbit Specific HRP/AEC (ABC) Detection IHC Kit, Abcam, Cambridge, UK). The following mouse or rabbit primary antibodies were used: anti-*PI-9* (mouse monoclonal, clone 7D8, 1:20, Santa Cruz Biotechnology Inc., Heidelberg, Germany), anti-*GrB* (rabbit polyclonal, 1:300, Abcam), anti-leukocyte-common antigen (CD45, rabbit polyclonal, 1:400, Santa Cruz Biotechnology Inc.), anti-cluster of differentiation 34 (CD34, mouse monoclonal, clone QBEnd/10, 1:100, Novocastra Laboratories, Newcastle upon Tyne, UK), and anti-human leukocyte antigen (HLR)-DR (mouse monoclonal, clone B8.12.2, 1:200, Immunotech, Marseilles, France).

### 2.3. Indirect Immunofluorescence of Imprints

Cell imprints were stained with a single antibody as described previously [22]. After 1 min fixation in cold acetone, membranes with harvested cells were washed in phosphate-buffered saline (PBS) and permeabilized in 0.2% Triton X-100. After washing, the imprints were incubated with the primary antibody (anti-*PI-9* or anti-*GrB* was diluted in 0.1% bovine serum albumin (BSA) in PBS) for 1 h at room temperature. Then, the specimens were rinsed three times in PBS and incubated with the secondary antibody (anti-mouse immunoglobulin G (IgG), 1:350, anti-rabbit IgG, 1:200, both fluorescein isothiocyanate-conjugated, Jackson ImmunoResearch Laboratories, West Grove, PA, USA) for 1 h at room temperature. After rinsing in PBS, the cells were mounted with Vectashield-propidium iodide (Vector Laboratories, Burlingame, CA, USA) to counterstain nuclear DNA.

### 2.4. Immunohistochemistry Assessment

The samples were examined using Olympus BX51 microscope (Olympus Co., Tokyo, Japan) at 200–1000× magnification, and images were recorded using a Vosskühler VDS CCD-1300 camera (VDS Vosskühler GmbH, Osnabrück, Germany), a Jenoptik ProgRes C12 plus camera (Jenoptik AG, Jena, Germany), and NIS Elements photo software (Laboratory Imaging, Prague, Czech Republic). The proportion of positive cells (expressed as a percentage) was calculated by two investigators (S.R.M. and K.J.) in the healthy human cornea, limbus, and adjacent perilimbal conjunctiva. The staining intensity was graded using a scale: N—no discernible staining; 1—weak staining; 2—moderate staining; and 3—intense staining. In the pathological corneas, we evaluated infiltrates and individual infiltrating cells within the stroma or endothelium separately for all detected markers: +++, ++, and + indicate >50%, 25–50%, and 1–25% positive cells, respectively.

### 2.5. Quantitative Real-Time PCR (qRT-PCR)

The expression of genes for *PI-9* and *GrB* genes in the corneal epithelium, endothelium, and conjunctiva was determined by qRT-PCR. A total RNA was isolated using ToRNAzol reagent (Gene Age Technologies, Prague, Czech Republic) with Polyacryl Carrier (Molecular Research Center, Cincinnati, OH, USA) according to instructions of the manufacturer. Total RNA (1 µg) was treated using deoxyribonuclease I (Promega, Madison, WI, USA) and then used for reverse transcription. Moloney murine leukemia virus (M-MLV) reverse transcriptase and random primers obtained from Promega in a total reaction volume of 25 µL were used for the synthesis of the first-strand complementary cDNA.

qRT-PCR was performed in a StepOne Plus Cycler (Applied Biosystems, Warrington, UK), using SYBR Green Master Mix (Applied Biosystems) following the instructions of the manufacturer. The primers used for the amplification of the glyceraldehyde-3-phosphate dehydrogenase (GAPDH), *GrB*, and *PI-9* genes were obtained from Generi Biotech (Hradec Kralove, Czech Republic): GAPDH: 5′-agccacatcgctcagacac-3′ (left), 5′-gcccaatacgaccaaatcc-3′, *GrB*: 5′-agatgcaaccaatcctgctt-3’(left), 5′-catgtcccccgatgatct-3′, *PI-9*: 5′-gaaacaccgcaacccagat-3′ (left), and 5′-actggaaagcccgatgaat-3′. The PCR parameters included denaturation at 95 °C for 3 min, then 40 cycles at 95 °C for 20 s, annealing at 60 °C for 30 s, and elongation at 72 °C for 30 s. Fluorescence data were collected at each cycle after an elongation step at 80 °C for 5 s and were analyzed on a StepOnePlus detection system (Applied Biosystems). A relative quantification model was used to calculate the expression of the target gene in comparison to GAPDH used as the endogenous control. Every sample was run in duplicate. At least four samples per group and two independent experiments were performed.

### 2.6. Tissue Culture

To study the effect of proinflammatory cytokines on *PI-9* and *GrB* gene expression, central cornea samples were divided into four quarters and cultured in RPMI 1640 medium (Sigma-Aldrich, St. Louis, MO, USA) containing 10% fetal calf serum (Sigma-Aldrich) and antibiotics (penicillin and streptomycin) non-stimulated or stimulated with human IFN-γ and TNF-α (PeproTech, Rocky Hill, NJ, USA). Both cytokines were used in a final concentration of 10 ng/mL; the concentration was selected on the basis of preliminary experiments. After 48 h cultivation, total mRNA was isolated and analyzed by qRT-PCR. Differences between the two groups were analyzed by Student’s *t*-test. A value of *p* ≤ 0.05 was considered statistically significant.

## 3. Results

### 3.1. Localization of PI-9 and GrB Proteins in Control Cornea and Conjunctiva

*PI-9* protein was detected in all tested tissues, particularly in the superficial and suprabasal epithelial layers and in the endothelial cells (Figure 1A, Table 1). *PI-9* was not found in the corneal stroma, with the exception of a few positive cells, and was sporadically detected in the peripheral part of the anterior stroma just beneath Bowman’s layer. Rare *PI-9*-positive cells were also found in the anterior stroma of the limbus and conjunctiva. All superficial layer imprints had a similar number of *PI-9*-positive cells (43%, 60%, and 65% of the endothelial, corneal epithelial, and conjunctival epithelial cells, respectively) (Figure 1B).

*GrB* was localized in the cornea, limbus, and conjunctival epithelium with predominant staining in the superficial cell layers (70% of the superficial cells). Only a weak signal was found in endothelial cells (1–5%), while an interrupted positive posterior line was observed in Descemet’s membrane in all control specimens. *GrB* was not detected in the corneal stroma, but some *GrB*-positive cells were present in the stromal part under the limbus and conjunctiva of all controls (Figure 1A). Abundant immunostaining was present in the human tonsils and testes used as positive controls for *PI-9* and *GrB*, respectively (Figure 1A). The corneal and conjunctival epithelial imprints showed *GrB* cytoplasmic positivity in approximately 40% of the cells. Scattered *GrB*-positive cells (1–5%) were present throughout the endothelium (Figure 1B).

The cells positive for HLA-DR antigen were detected in the peripheral corneal, limbal, and perilimbal conjunctival epithelium; we found sporadic positivity in the central corneal stroma, and numerous HLA-DR-positive cells were observed in the stroma of the peripheral cornea, limbus, and conjunctiva of all control corneas. CD34 was present in keratocytes, and rarely, a few CD45-positive cells were located at the periphery of the corneal stroma.

### 3.2. Expression of PI-9 and GrB Genes in Control and Cytokine-Stimulated Corneas

*PI-9* and *GrB* mRNA expression in the control human corneas was confirmed using qRT-PCR. The expression of the *PI-9* gene was apparently higher in the corneal epithelium than in the corneal endothelium, but due to the individual variability of samples (a wide SD) this difference could not be proven statistically significant. *GrB* was, in accordance with immunohistochemistry results, only negligibly detected in corneal endothelium, but highly expressed in the corneal epithelium (Figure 2A). The expression of both genes was significantly enhanced when the corneas were cultured with the proinflammatory molecules IFN-γ γ (10 ng/mL) and TNF-α γ (10 ng/mL) (Figure 2B). Besides the *PI-9* and *GrB* mRNAs, the expression of genes for other proinflammatory molecules, such as inducible nitric oxide synthase, interleukin 1 (IL)-1α, or IL-6, were significantly enhanced (data not shown).

### 3.3. The Localization of PI-9 and GrB in Pathological Explants

*PI-9* expression in the epithelium and endothelium of corneas with melting or herpes virus keratitis exhibited wide diversity ranging from negative to moderate positive staining in almost all cells (Table 2). Compared with the completely *PI-9*-negative central stroma of all control corneas, abundant *PI-9* expression was found in numerous individual cells in the stroma of the pathological specimens. Besides these cells, *PI-9*-positive stromal infiltrates were found in 40% of explants. A massive infiltration of *PI-9*-positive cells was observed within the endothelium of one melted (M) graft (Figure 3, M5).

*GrB* expression in the epithelium and endothelium of the pathological corneas exhibited wide diversity. Within the corneal stroma, we detected a similar *GrB* immunostaining pattern to that of *PI-9* (Table 3).

HLA-DR and CD45 were massively expressed in individual stromal cells and cells present in stromal infiltrates (Figure 3). CD34 immunostaining was absent or weak in stromal infiltrates (Figure 3). In general, the CD34 signal was decreased at locations where positive immunostaining for *PI-9*, *GrB*, HLA-DR, and CD45 was observed.

The epithelium of the melted corneas was significantly damaged or even totally absent (Figure 3, M3). On the other hand, stromal infiltrates in the corneas with herpes virus keratitis were predominantly found in the anterior part of the stroma, just beneath the basement membrane of the epithelium (Figure 3, K1).

## 4. Discussion

We detected *GrB* and its related inhibitor *PI-9* in healthy human cornea, limbus, and conjunctiva. The abnormal expression of both proteins was found in the stroma of melted and herpetic explants. In healthy cornea, *PI-9* was predominantly present in the lining layers (superficial epithelium and the endothelium), where its localization was consistent with its defense role against *GrB* released during local immune response [15]. *GrB* had similar localization but its expression, particularly in the endothelium, was much lower, as shown using immunohistochemistry and confirmed by qRT-PCR. The presence of *PI-9* in the corneal endothelium is also consistent with its abundant expression in the pleural and peritoneal mesothelium, which share some phenotypic similarity with the corneal endothelium, e.g., immunostaining for typical mesothelial proteins such mesothelin, Hector Battifora mesothelial-1 (HBME-1), and calbindin 2 [9,22]. *PI-9* expression in the cornea may also protect cells from apoptosis induced by *GrB*-positive immune cells, e.g., cytotoxic lymphocytes that enter the cornea from the surrounding tissue [10,15].

We have shown that proinflammatory molecules IFN-**γ** and TNF-α stimulate *PI-9* and *GrB* gene expression in cultured corneas, similarly to their stimulation of the expression of other proinflammatory molecules such as nitric oxide synthase, IL-1α and IL-6. Since IFN-**γ** and TNF-α are cytokines involved in inflammatory reactions and in graft rejection, and the expression of *PI-9* and *GrB* is regulated by these cytokines, we suggest that the pathway *PI-9*/*GrB* can be involved in regulation of immunity in the eye and in the maintenance of the immunological privilege of the cornea. We suppose that during corneal rejection, *PI-9* can inhibit the cytotoxic effect of *GrB* possessed by immunologically active cells such as dendritic cells, macrophages, and lymphocytes participating in corneal rejection [23], and in this way protect corneal cells from *GrB*-induced apoptosis [24].

As both *PI-9* and *GrB* are molecules involved in immune response, we have studied their potential involvement in pathological corneas with herpes virus keratitis, where immune response against herpes simplex virus represents both innate and adaptive immune mechanisms [20], and in melted corneas, where infiltrates of inflammatory cells are found [21]. Compared with the control tissues, *PI-9* and Gr-B immunostaining in the pathological tissues was more heterogeneous, showing both higher and lower positivity in the corneal epithelium and endothelium of the individual specimens. This heterogeneity could reflect the disease stage at which the explants were collected. We found distinctly *PI-9*-positive infiltrates within both melted and herpetic explants. The intense *PI-9*, *GrB*, HLA-DR, and CD45 staining in individual stromal cell infiltrates indicates that these positive cells migrated to the cornea during the pathological process and represent cells of the immune system, particularly macrophages, dendritic cells, B and T lymphocytes, and NK cells. The concurrent significant decrease in the keratocyte marker CD34 at locations with increased positivity for *PI-9*, *GrB*, HLA-DR, and CD45 may be related to keratocyte differentiation into fibroblasts or myofibroblasts [25,26]. The limitation of our study is the small number of pathological corneas analyzed, thus the exact function of *PI-9* has to be clarified in other studies.

The increased *GrB* signal in the pathological stroma, particularly that of melted tissue, is consistent with its capacity to degrade proteoglycans, fibronectin, laminin, and other components of the corneal extracellular matrix [27,28]. The increased presence of *PI-9* in pathological corneas may provide protection against *GrB* released from neighboring cytotoxic lymphocytes, as has been shown during an immune cytotoxic attack [6,9]. The defense function of *PI-9* against a *GrB* cytotoxic reaction has also been described in renal rejection, where the abundant presence of *PI-9* in tubular epithelial cells in subclinical, but not acute rejection, correlated with the presence of *GrB*-positive cytotoxic T lymphocytes in the graft [29]. On the other hand, *PI-9* mRNA expression declined in patients who underwent allogeneic hematopoietic stem cell transplantation. These findings indicate that the properly timed synthesis of *PI-9* may be crucial for protecting cells against destruction, and thus it participates in the modulation of immune response [30].

In conclusion, we localized for the first time the *PI-9* and *GrB* proteins in healthy human cornea, showed their ability to be stimulated by proinflammatory cytokines, and confirmed their participation in immune response in pathological corneas.

## Figures and Tables

**Figure 1 biology-11-00793-f001:**
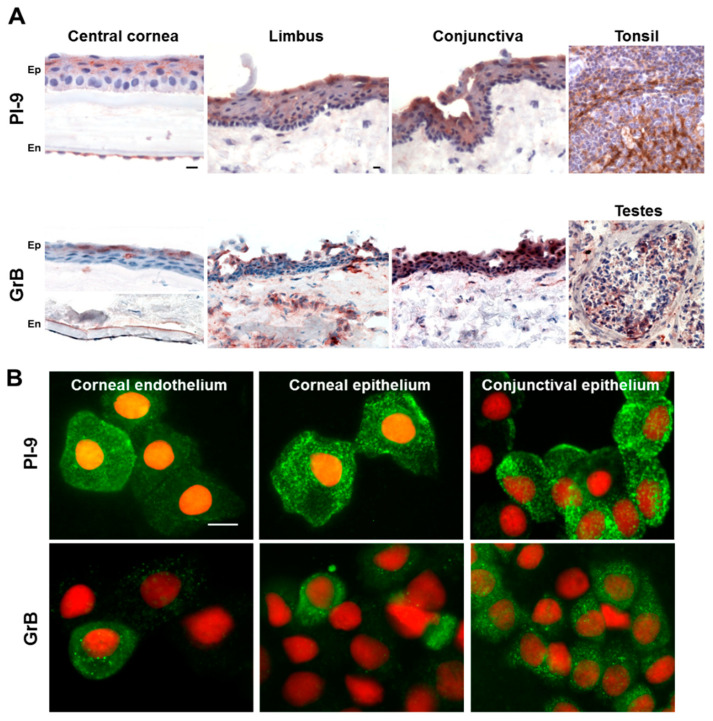
Detection of *PI-9* and *GrB* in the control human cornea, limbus, and conjunctiva. (**A**) Enzymatic immunohistochemical detection of *PI-9* and *GrB* in control human cornea (Ep: epithelium and En: endothelium), limbus, and conjunctiva. Human tonsil and testes were used as positive controls for *PI-9* and *GrB*, respectively. (**B**) Detection of *PI-9* and *GrB* in the control human cornea using indirect fluorescent immunocytochemistry. Corneal endothelium, epithelium, and conjunctival epithelium show positive intracytoplasmic dots. Nuclei were stained with propidium iodide. Scale bars = 10 μm.

**Figure 2 biology-11-00793-f002:**
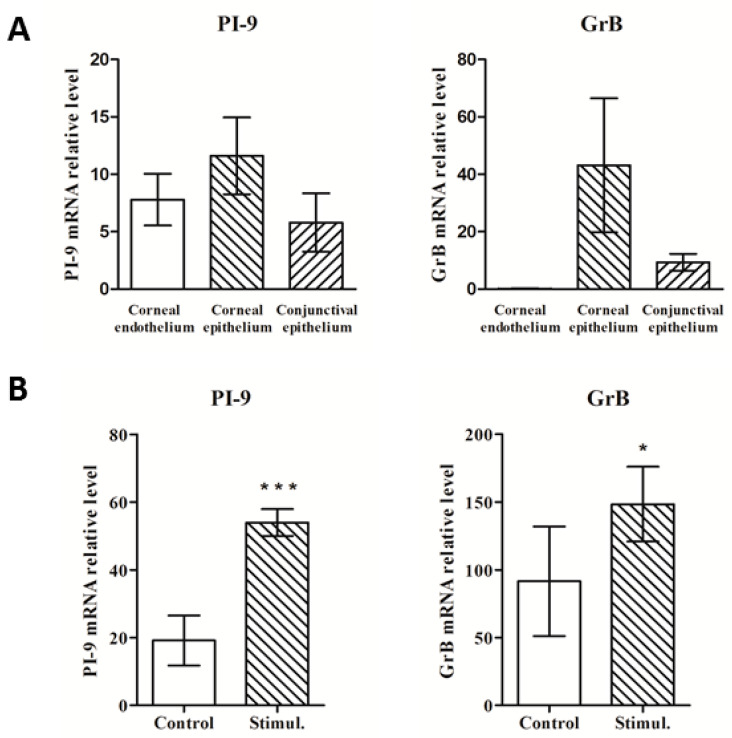
Expression of the *PI-9* and *GrB* genes in human corneas. (**A**) *PI-9* and *GrB* gene expression in the corneal epithelium, endothelium, and conjunctiva. (**B**) Enhanced *PI-9* and *GrB* gene expression in corneas cultured in medium (control) or in medium with proinflammatory cytokines (Stimul.). Bars represent ± SD from five individual corneas. * *p* ≤ 0.05 and *** *p* ≤ 0.001 indicate statistical significance as compared with the control.

**Figure 3 biology-11-00793-f003:**
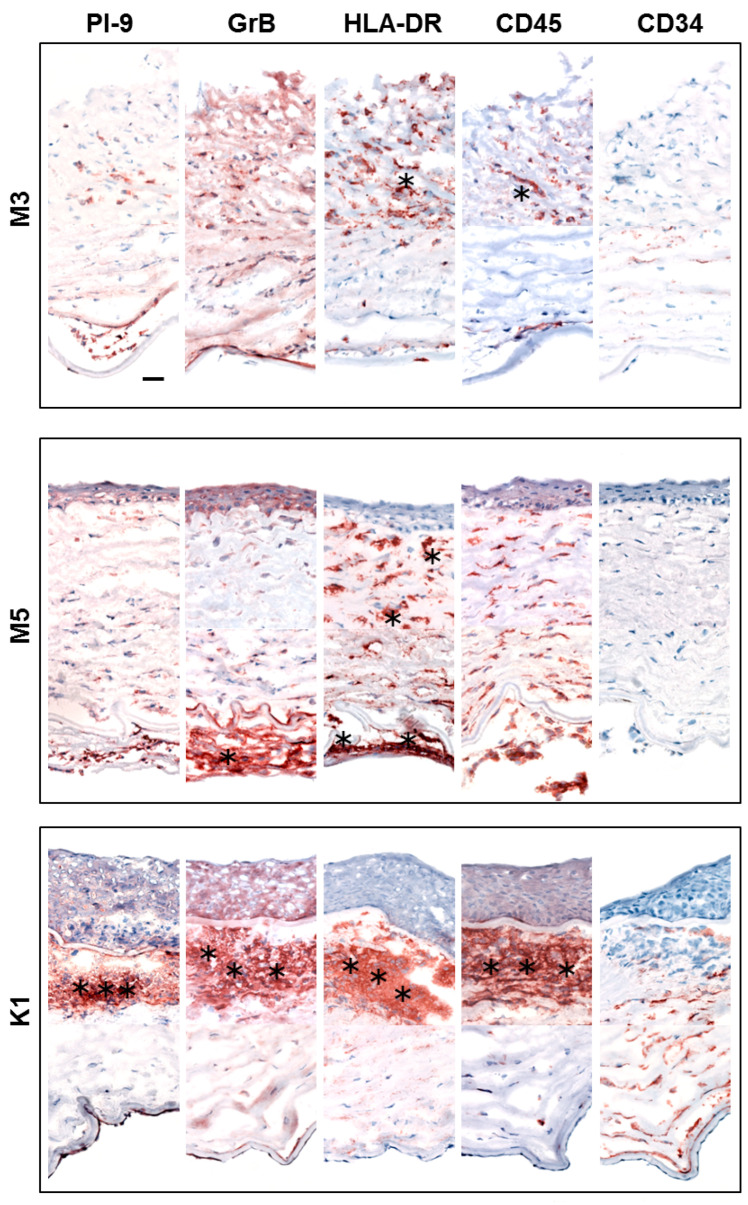
Enzymatic immunohistochemical detection of *PI-9*, *GrB*, HLA-DR, CD45, and CD34 in melted corneas (**M3** and **M5**) and in the cornea with herpes virus keratitis (**K1**). Some examples of infiltrates are indicated by asterisks. (*). Scale bar = 50 μm.

**Table 1 biology-11-00793-t001:** The presence of *PI-9* in control (Co) human corneas and conjunctiva.

Sample	Corneal Endothelium	Corneal Stroma	Corneal Epithelium			Limbus			Conjunctiva		
			B	SB	S	B	SB	S	B	SB	S
	Percentage of positive cells/intensity										
Co1	60/2	N	N	7/1	43/2	7/1	10/1	40/1	7/1	10/1	45/1
Co2	67/2	N	13/1	37/1	53/2	10/1	40/2	35/2	10/1	45/2	40/2
Co3	50/1	N	7/1	43/1	73/2	N	40/1	60/2	N	40/1	60/2
Co4	10/1	N	N	37/1	53/2	10/1	45/1	55/2	10/1	50/1	60/2
Co5	60/2	N	N	35/2	45/1	N	60/2	45/2	N	65/2	45/2

Number of cells stained: % (percentage)/staining intensity: N, no staining; 1, weak; 2, moderate; and 3, intense. B: basal, SB: suprabasal, and S: superficial layer of the epithelium.

**Table 2 biology-11-00793-t002:** The presence of *PI-9* immunostaining in melted corneas (M) and corneas with herpes virus keratitis (K).

Pathology	Corneal Endothelium	Corneal Epithelium	Corneal Stroma	
			Infiltrates	Individual cells
M1	N	15/1	nc	+
M2	35/1	20/1	nc	++
M3	np	np	++	++
M4 *	45/1	45/1	+	+
M5 *	90/3	35/1	nc	++
K1	90/3	1/1	++	+
K2	N	20/1	+	+
K3	75/2	60/2	nc	+
K4	80/2	30/1	nc	+
K5	np	50/1	nc	+
K6	50/1	30/1	++	++
K7	25/1	N	nc	+
K8	35/1	N	nc	N

Number of endothelial and epithelial cells stained: % (percentage)/staining intensity: N, no staining; 1, weak; 2, moderate; and 3, intense. Infiltrates and individual cells in the stroma were evaluated separately: +++, ++, and + indicate >50%, 25–50%, and 1–25% positive cells, respectively; nc: cells were not present. * Melted corneal grafts.

**Table 3 biology-11-00793-t003:** The enzymatic immunohistochemical detection of *GrB*, HLA-DR, CD45, and CD34 in the corneal stroma of melted (M) corneas and corneas with herpes virus keratitis (K).

Pathology	Corneal Stroma							
	Infiltrates				Individual Cells			
	*GrB*	HLA-DR	CD45	CD34	*GrB*	HLA-DR	CD45	CD34
M1	nc	nc	nc	nc	+	+	+	+++
M2	nc	nc	nc	nc	+	++	++	+++
M3	++	+++	++	N	+	+++	++	+
M4 *	++	++	+++	N	++	+++	++	++
M5 *	nc	nc	nc	nc	++	+++	+++	N
K1	+++	++	+++	N	+	+	+	+++
K2	+	++	++	+	+	++	+	++
K3	nc	nc	nc	nc	+	++	+	+++
K4	nc	nc	nc	nc	+	+++	+	++
K5	nc	nc	nc	nc	+	+	+	+
K6	nc	++	nc	nc	+	++	+	+++
K7	++	++	+	nc	+	++	+	+++
K8	nc	nc	nc	nc	+	++	+	++

Infiltrates and individual cells within the stroma were evaluated separately: +++, ++, and + indicate >50%, 25–50%, and 1–25% positive cells, respectively; N indicates no staining; nc: cells were not present. * Melted corneal grafts.

## Data Availability

Not applicable.

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
