# Peer review of "Presence of Protease Inhibitor 9 and Granzyme B in Healthy and Pathological Human Corneas"

_biology, 2022, doi:10.3390/biology11050793_

Round 1

Reviewer 1 Report

In this manuscript, the authors determine the presence and expression of protease inhibitor 9 (PI-9) and granzyme B (GrB) in healthy and pathological human corneas and suggest that the participation during inflammatory response. However, I encourage the author to address few concerns to further strengthen for acceptation in the journal:

  1. Materials and Methods; page 3 line 133.. 1-weak staining; 2-moderate staining; and 3-intense staining… this scale is very subjective, the assignation of value was interpretation by only one investigator?, I suggest the use of analysis software such as fiji (ImageJ), that would be express the pixels/area and would allow establishment the range for the scale
  2. In the figure 1b include the magnification. Apparently, the magnification in PI-9 and GrB is different
  3. Results;
    1. Page 6 line 203-207: HLA-DR-positive cells were detected in the peripheral corneal, limbal, and perilimbal conjunctival epithelium……………CD45-positive cells were located in the corneal stroma periphery. In this case the description not included any figure or data not show.
    2. Page 6 line 209:…. which showed higher expression in the corneal epithelium compared to the endothelium (Figure 2A)...In the graph of expression of mRNA PI-9 y GzB not show significative differences between tissues, then the asseveration is wrong
    3. Page 9 line 255: Figure 3 include the magnification and I suggest indicated with arrows the Infiltrates

Author Response

Reviewer 1

In this manuscript, the authors determine the presence and expression of protease inhibitor 9 (PI-9) and granzyme B (GrB) in healthy and pathological human corneas and suggest that the participation during inflammatory response. However, I encourage the author to address few concerns to further strengthen for acceptation in the journal:

3.Materials and Methods; page 3 line 133.. 1-weak staining; 2-moderate staining; and 3-intense staining… this scale is very subjective, the assignation of value was interpretation by only one investigator?, I suggest the use of analysis software such as fiji (ImageJ), that would be express the pixels/area and would allow establishment the range for the scale.

Indeed, the attribution of the intensity scale is in this case subjective. However, as it is used only as a relatively weak correction factor relative to the percentage evaluation, which is the major evaluator, we did not consider it necessary to make the evaluation more accurate. Moreover, it should be taken in account that even rigorous intensity profile and value determination would still be affected by other factors related to the staining procedure specimen variations and parameters of the image acquisition which are not always simple to control. Therefore, we consider this coarse division into three categories acceptable. The specimens were assessed by two evaluators, and it was indicated in Materials and Methods.

4.In the figure 1b include the magnification. Apparently, the magnification in PI-9 and GrB is different

Yes, that is true, but it should be noted that there are two scale bars in the Figure 1. One in the top right image and the second in adjacent one, which is than valid for all other images in the panel as indicated in the legend. The aim to use different scales was to show clearly the positivity in different layers of the corneal epithelium.

  1. Results;
  2. Page 6 line 203-207: HLA-DR-positive cells were detected in the peripheral corneal, limbal, and perilimbal conjunctival epithelium……………CD45-positive cells were located in the corneal stroma periphery. In this case the description not included any figure or data not show.

The whole sentence is more precise: CD34 was present in keratocytes, and only few CD45-positive cells were located in the corneal stroma periphery. As the only few (i.e. less than 1%) and just on specific location (periphery of the corneal stroma, i.e. location where corneal stroma is changed to conjunctival stoma), we did not feel the need to specify this information. To be absolutely clear the sentence was changed as follows: CD34 was present in keratocytes, and rarely few CD45-positive cells were located at the periphery of the corneal stroma. 

  1. Page 6 line 209:…. which showed higher expression in the corneal epithelium compared to the endothelium (Figure 2A)...In the graph of expression of mRNA PI-9 y GzB not show significative differences between tissues, then the asseveration is wrong

The reviewer is right, it was changed as suggested: The expression of the PI-9 gene was apparently higher in the corneal epithelium than in corneal endothelium, but from the reason of individual variability of samples (a wide SD) this difference could not be proven statistically significant. GrB was, in the accordance with immunohistochemistry results, only negligibly detected in corneal endothelium, but highly expressed in the corneal epithelium (Figure 2A).    

  1. Page 9 line 255: Figure 3 include the magnification and I suggest indicated with arrows the Infiltrates

Scale bar for all images of Figure 3 is present in the top right image. The examples of cell infiltrates are indicated by asterisks (*).

Reviewer 2 Report

I find the study and subsequent finding of this research very interesting, although, I would like to find a paragraph in the discussion that explains its implication and/or clinical application at a practical level.

Author Response

I find the study and subsequent finding of this research very interesting, although, I would like to find a paragraph in the discussion that explains its implication and/or clinical application at a practical level.

More information regarding potential mechanism of PI-9 and GrB during corneal pathologies was added in the Discussion section. On the other hand, additional experiments will be necessary to confirm the suggested mechanisms.

Reviewer 3 Report

The authors in this report have studied the role of PI-9 and GrB proteins in healthy and infected corneas and focus on their participation in immune response in pathological corneas. Although the study claims to target pathological candidates but there are some missing links.

  1. Authors describe they found the localization of PI-9 and grb in naïve corneas. What is the source they are coming from?

  1. In discussion, authors mention IFN-g and TNF-a stimulate PI-9 and GrB gene expression in cultured corneas, this could be from a wide variety of pro-inflammatory cytokines like the authors mentioned IL-1a or IL-6 but a complete assay analysis involving a bunch of cytokines/chemokines needs to be studied.

  1. Another confusion is the PI-9/GrB pathway that authors suggest is involved in regulation of immunity in the eye, the mechanism is not clear.

  1. The authors also states that cell infiltration into the cornea namely macrophages, dendritic cells etc. what is the percentage of these cell infiltrates? That is important in clinical terms.

Author Response

  1. Authors describe they found the localization of PI-9 and grb in naïve corneas. What is the source they are coming from?

The reviewer is right, “native” is not a suitable word, it was misused to distinguish these corneas from those ones that have been preserved in hypothermic conditions. Word native was deleted from the manuscript. Thank you for this comment.

  1. In discussion, authors mention IFN-g and TNF-a stimulate PI-9 and GrB gene expression in cultured corneas, this could be from a wide variety of pro-inflammatory cytokines like the authors mentioned IL-1a or IL-6 but a complete assay analysis involving a bunch of cytokines/chemokines needs to be studied.

In this case, pro-inflammatory factors (IFN-γ and TNF-α) were used to stimulate inflammation under experimental conditions. Therefore, we were not interested in comparisons between individual anti-inflammatory cytokines, but just wanted to demonstrate that an inflammation (even stimulated) increases the expression of both proteins (PI-9 and GrB). The discussion is related to pathological conditions under which the expression of PI-9 and GrB is changed.

  1. Another confusion is the PI-9/GrB pathway that authors suggest is involved in regulation of immunity in the eye, the mechanism is not clear.

We have shown that proinflammatory cytokines molecules IFN-γ and TNF-α stimulate PI-9 and GrB gene expression in cultured corneas, similarly to their stimulation of the expression of other proinflammatory molecules such as nitric oxide synthase, IL 1α and IL-6. Since IFN-γ and TNF-α are cytokines involved in inflammatory reactions and in graft rejection, and the expression of PI-9 and GrB is regulated by these cytokines, we suggest that the pathway PI-9/GrB can be involved in regulation of immunity in the eye and in maintenance of the immunological privilege of the cornea.

To the concerned text present in Discussion we have provided more detailed explanations: We suppose that during corneal rejection PI-9 can inhibit cytotoxic effect of GrB possessed by immunologically active cells such as dendritic cells, macrophages and lymphocytes participating on corneal rejection (PMID: 32354200), and by this way to protect corneal cells from GrB induced apoptosis (PMID: 24918613). When Pi-9 is depleted, GrB is not blocked and may trigger the cell death process. Similar situation can occur in melted corneas (melted corneal grafts) or corneas from patients suffering from herpes virus keratitis. This hypothesis is consistent with the heterogenous presence (percentages) of Pi-9 in diseased corneas. The limitation of our study is the small number of pathological corneas analyzed, thus the exact function of PI-9 have to be clarified in other studies.

Some parts of this text were included in the Discussion section.

  1. The authors also states that cell infiltration into the cornea namely macrophages, dendritic cells etc. what is the percentage of these cell infiltrates? That is important in clinical terms.

These results are shown in Table 3 (please see also Table 3 legend for percentual range). The presence of some of the markers typical for dendritic cells (HLA-DR), macrophages (CD45) (i.e. both infiltrating cells), and for CD34-positive corneal stromal cells (keratocytes) is shown. While to determine the accurate percentage of positive cells for various markers in control samples was possible, it was not feasible in pathological corneas due to the high cell concentration.

Round 2

Reviewer 3 Report

Did the authors do a dose dependent study for IFN-g and TNF-a for tissue culture experiments? How did they decide the dose? Also, please mention the dose used in the results section as well.

Author Response

Did the authors do a dose dependent study for IFN-g and TNF-a for tissue culture experiments? How did they decide the dose? Also, please mention the dose used in the results section as well.

The concentration of proinflammatory cytokines used for the in vitro stimulation was 10 ng/mL. This concentration was selected on the basis of preliminary experiments and on the basis of our published data. As Reviewer required, we added this information in the revised version (page 4, l.164-167, page 6, l. 218).